# Universal coverage but unmet need: National and regional estimates of attrition across the diabetes care continuum in Thailand

**Lily D. Yan**[1], **Piya Hanvoravongchai**[2], **Wichai Aekplakorn**[3], **Suwat Chariyalertsak**[4,5], **Pattapong Kessomboon**[6], **Sawitri Assanangkornchai**[7], **Surasak Taneepanichskul**[8], **Nareemarn Neelapaichit**[9,10], **Andrew C. Stokes**[11]*

**1** Internal Medicine, Boston Medical Center, Boston, MA, United States of America, **2** Faculty of Medicine, Chulalongkorn University, Bangkok, Thailand, **3** Department of Community Medicine, Faculty of Medicine Ramathibodi Hospital, Mahidol University, Bangkok, Thailand, **4** Research Institute for Health Sciences, Chiang Mai University, Chiang Mai, Thailand, **5** Faculty of Public Health, Chiang Mai University, Chiang Mai, Thailand, **6** Medicine, Khon Kaen University, Khon Kaen, Thailand, **7** Epidemiology Unit, Prince of Songkla University, Songkhla, Thailand, **8** College of Public Health Science, Chulalongkorn University, Thailand, **9** Ramathibodi School of Nursing, Faculty of Medicine Ramathibodi Hospital, Mahidol University, Bangkok, Thailand, **10** Thailand Research Institute for Health Sciences, Chiang Mai University, Chiang Mai, Thailand, **11** Global Health, Boston University School of Public Health, Boston, MA, United States of America

* acstokes@bu.edu

## Abstract

### Background

Diabetes is a growing challenge in Thailand. Data to assess health system response to diabetes is scarce. We assessed what factors influence diabetes care cascade retention, under universal health coverage.

### Methods

We conducted a cross-sectional analysis of the 2014 Thai National Health Examination Survey. Diabetes was defined as fasting plasma glucose $\geq$126mg/dL or on treatment. National and regional care cascades were constructed across screening, diagnosis, treatment, and control. Unmet need was defined as the total loss across cascade levels. Logistic regression was used to examine the demographic and healthcare factors associated with cascade attrition.

### Findings

We included 15,663 individuals. Among Thai adults aged 20+ with diabetes, 67.0% (95% CI 60.9% to 73.1%) were screened, 34.0% (95% CI 30.6% to 37.2%) were diagnosed, 33.3% (95% CI 29.9% to 36.7%) were treated, and 26.0% (95% CI 22.9% to 29.1%) were controlled. Total unmet need was 74.0% (95% CI 70.9% to 77.1%), with regional variation ranging from 58.4% (95% CI 45.0% to 71.8%) in South to 78.0% (95% CI 73.0% to 83.0%) in Northeast. Multivariable models indicated older age (OR 1.76), males (OR 0.65), and a higher density of medical staff (OR 2.40) and health centers (OR 1.58) were significantly associated with being diagnosed among people with diabetes. Older age (OR 1.80) and

**Data Availability Statement:** Data used in this study are not freely available because of restrictions involving potentially identifying information on surveyed participants, including name, address, birthday, province, and extensive medical history through self report, physical exam. Requests for data access would be subject to scrutiny by researchers from Mahidol University. Access to data will only be granted after agreement from Mahidol University. For further inquires, please contact Dr. Wichai Aekplakorn (wichai. aek@mahidol.ac.th), Dr. Piya Hangvoravongchai (piya.h@chula.ac.th), or the Health System Research institute (hsri@hsri.or.th).

**Funding:** The authors received no specific funding for this work.

**Competing interests:** The authors have declared that no competing interests exist.

**Abbreviations:** BMI, body mass index; CRL, continuation ratio logit; LMIC, low and middle income country; NCD, noncommunicable diseases; NHES, national health examination survey; OR, odds ratio; UHC, universal health coverage.

higher geographical density of medical staff (OR 1.82) and health centers (OR 1.56) were significantly associated with being controlled.

## Conclusions

Substantial attrition in the diabetes care continuum was observed at diabetes screening and diagnosis, related to both individual and health system factors. Even with universal health insurance, Thailand still needs effective behavioral and structural interventions, especially in primary health care settings, to address unmet need in diabetes care for its population.

## Introduction

The global burden of non-communicable disease (NCDs) has grown substantially in recent years, with the most rapid increase occurring in low- and middle-income countries (LMICs) [1]. This shifting landscape poses a significant challenge to health care systems, particularly in LMIC settings with limited infrastructure for addressing complex diseases such as diabetes which require coordinated care and long-term management. Globally, the prevalence of diabetes in adults has increased in every country since 1980, with the burden increasing most rapidly in LMICs [2]. The age standardized prevalence of diabetes globally has increased from four to nine percent for men, and from five to eight percent for women, which in absolute terms translates into an increase of 314 million more people with diabetes worldwide over the past 40 years [2]. Diabetes accounts for more than two million deaths a year, and is the seventh leading cause of disability worldwide [3].

Thailand has recently transitioned from a low middle income country to a high middle income country, with a GDP that more than tripled between 2000 and 2017, and is in the midst of an epidemiologic transition [4]. Chronic diseases were estimated to account for 74% of all deaths nationwide in 2016 [5]. By disease burden, diabetes was the leading cause for men, and the seventh cause for women in 2014 [6]. The age adjusted prevalence of diabetes increased from eight to ten percent from 2004 to 2014 in Thailand [7].

Thailand was one of the earliest LMIC to implement universal health insurance coverage in 2002, with over 99% of the population covered by one of three major insurance schemes [4]. With the growing burden of diabetes, expanded health systems under universal health coverage must also provide efficient and high quality care, as increased quantity of care alone has not resulted in healthier populations, satisfied patients, or equity of outcomes [8,9]. Global standards around high quality healthcare in LMIC, including competent care, patient experience, health outcomes, and confidence in the system, remain markedly undeveloped [8]. Furthermore, there are a lack of data on where best to intervene to strengthen health systems to respond in particular to chronic diseases like diabetes [9].

One promising approach to addressing this challenge is the use of care cascades to identify points of loss, or gaps, in the chronic disease care continuum across screening, diagnosis, treatment, and control. Care cascades were originally used to model loss to follow up in HIV/AIDS care and have subsequently been applied to a range of other chronic conditions, including hypertension and diabetes [10,11]. Prior studies using data across multiple LMIC, for example, revealed large gaps between screening and diagnosis, with 80% unmet need in diabetes care [12–14].

In the present study, we estimate national and regional levels of unmet need for care across the diabetes care continuum in Thailand using data from the Thai National Health

Examination Survey (NHES-V). We hypothesize that sociodemographic and health-system factors both contribute to attrition across stages of the cascade, including diabetes screening, diagnosis, treatment, and control. The gaps identified by this approach may then be targets for future interventions to improve diabetes control, morbidity, and mortality in Thailand.

## Methods

### Design, setting

This study used the 2014 Thai National Health Examination survey (NHES V), the largest cross-sectional, noninstitutionalized population representative survey in Thailand, completed every five years. The survey utilizes four stage sampling: 1) five provinces randomly selected from each of five regions, 2) two to three districts randomly selected from each province, 3) 24 enumeration areas randomly selected from each district (with balance of urban and rural), and 4) individuals of both sexes from each age group randomly selected from each enumeration area.

Data collection was conducted through face to face interviews, with a physical exam portion that collected blood samples after overnight fasting for eight hours. Blood samples were transferred to provincial hospitals for fasting plasma glucose testing using an enzymatic hexokinase method. All provincial laboratories were standardized to the central laboratory at the Department of Medical Service, Ministry of Public Health. In 2014, there were a total of 22,095 participants aged ≥20 years, and 8.8% adults had available blood samples [7].

Health system factors (hospitals, health centers, healthcare providers, and public health nurses) were abstracted from annual reports by the Policy and Strategy Bureau, and include both public and private hospitals [15]. These data were merged with NHES by province, the second subnational administrative level.

### Participants

All adults aged 20 and older, with a fasting plasma glucose were included [16,17]. Participants with missing age, sex, religion, BMI or missing information on diabetes screening or diagnosis were excluded. We did not distinguish between type I and type II diabetes because care targets should not change based on the type of diabetes. A flow chart of study exclusions is presented in S1 Fig.

### Measurements

Diabetes was defined as a fasting plasma glucose ≥126mg/dL or on treatment for diabetes (oral glycemic medications in the last two weeks, insulin in the last two weeks, or lifestyle modification specifically for diabetes such as diet, exercise, or weight loss). Prediabetes was defined as anyone with a fasting plasma glucose ≥100 mg/dL and <126, and not on treatment. Normoglycemia was defined as fasting plasma glucose <100 and not on treatment.

For the care cascade, five mutually exclusive and exhaustive categories were created: 1) unscreened (fasting plasma glucose ≥126 mg/dL, never tested for high blood sugar or diabetes; no reported prior diagnosis) 2) screened, undiagnosed (fasting plasma glucose ≥126 mg/dL; reported being tested ever; no reported prior diagnosis of diabetes); 3) diagnosed, untreated (prior reported diagnosis of diabetes, but no reported current use of oral glycemic medication or insulin therapy or lifestyle modification); 4) treated, uncontrolled (reported current use of oral glycemic medication, insulin, or lifestyle modification with fasting plasma glucose ≥183 mg/dL); 5) treated, controlled (reported current use of diabetes medication or lifestyle modification with fasting plasma glucose <183 mg/dL). A fasting plasma glucose of 183 corresponds

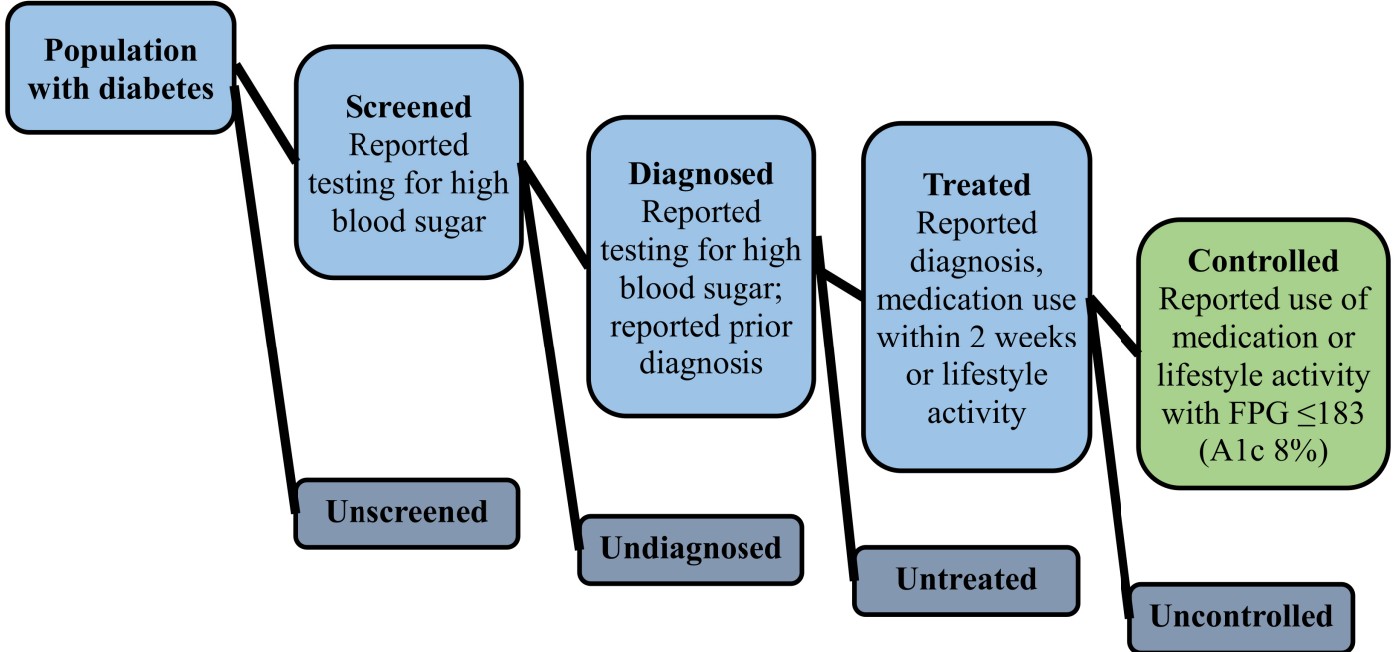

**Fig 1. Diabetes care cascade framework.**

to a HgbA1c 8% [14]. To examine loss across the cascade, we calculated the number of individuals with diabetes reaching each state of the cascade as a proportion of those reaching the prior stage. Unmet need was defined as the sum of first four categories (unscreened, undiagnosed, untreated, uncontrolled). Fig 1 presents a visual depiction of the cascade model used in the present study. Care cascades were constructed for 1) total population, and 2) stratified by region.

The main outcomes of interest in this study were people with diabetes who were screened, diagnosed, treated, and controlled. Independent variables included region, individual factors (age, sex, BMI, highest educational level) and health system factors (healthcare provider density, hospital density, health center density). For region, we combined data from Bangkok and Central, given the city is nested within the region and the relatively small sample size. Age was included as a continuous variable in 10 year increments. Information on family history and treatment adherence could not be included due to the amount of missing data. For each health system factor, we first calculated the ratio of population per factor (eg number of providers) by province, and then standardized these ratios so that a one unit increase in the model for that variable would represent a one standard deviation increase from the mean. Medical staff included physicians and nurses. Public health nurses are a special cadre of nurses not involved in direct clinical care, but instead lead public health initiatives. We hypothesized a combination of both individual and health system factors contributed to attrition across cascade levels.

## Statistical analysis

To examine this attrition, we performed multivariable modeling using un-nested logistic regression. Based on initial analysis of the diabetes care cascade in the present study, we modeled the following outcomes: 1) probability of screening conditional on diabetes; 2) probability

of diagnosis conditional on diabetes; 3) probability of control conditional on diabetes. We did not separately model treatment as the care cascade revealed nearly complete progression of the sample between diagnosis and treatment. In a secondary analysis, we used continuation ratio logit (CRL) regression, a method appropriate for modeling sequential processes in which outcomes are nested. A fully saturated CRL model is equivalent to a series of binary logit models implemented on each nested outcome. We modeled the following outcomes: 1) probability of screening conditional on diabetes; 2) probability of diagnosis conditional on screening; 3) probability of control conditional on diagnosis.

Survey data were weighted according to the inverse probability of being sampled based on the 2014 registered Thai population. The Thai 2010 Census was used for age standardization. All analyses were performed in Stata/SE 15.1 (StataCorp, College Station, TX). The family of svyset commands were used to adjust for survey weights [18].

### Ethics

This research study was approved by the Thai Ministry of Health and the institutional review board at Mahidol University, Bangkok, Thailand as well as the institutional review board at Boston University School of Public Health. A waiver of consent was granted as no identifiable patient information was included.

## Results

A total of 15,663 adults greater than 20 years were included in this analysis (Table 1). Participants tended to be middle aged (22.4% in 40–49 years, 24.0% in 50–59 years), female (52.4%), and normal BMI (53.4%). The majority of people had a primary education or less (57.7%), were overwhelming Buddhist (94.0%), and from rural areas (56.0%). About 17.3% of the sample was from Bangkok.

### Prevalence of diabetes

The age standardized prevalence of diabetes was 8.82% (95% CI 8.21% to 9.43%), and of prediabetes was 16.3% (95% CI 15.3 to 17.3) (Table 1, S1 Table). There was a higher prevalence of both prediabetes and diabetes in older age groups, and higher BMI categories. While males had a higher prevalence of prediabetes compared to females (17.9% vs 14.9%), they had a slightly lower prevalence of diabetes (8.28% vs 9.26%). Diabetes prevalence declined slightly with increasing educational levels (10.0% for primary or less vs 7.03% for university). Lastly, between the regions, diabetes was most prevalent in Central (10.8%), and least prevalent in South (6.11%).

### Diabetes care cascade

Fig 2 shows the diabetes care cascade. Among all people with diabetes, 67.0% (95% CI 60.9% to 73.1%) were screened, 34.0% (95% CI 30.6% to 37.4%) were diagnosed, 33.3% (95% CI 29.9% to 36.7%) were treated, and 26.0% (95% CI 22.9% to 29.1%) were controlled. The unmet need for diabetes was 74.0% (95% CI 70.9% to 77.1%). The largest gaps occurred at screening and diagnosis, while most people with diabetes who were diagnosed were on either lifestyle or medication treatment. In another way to examine this data, among the total population, approximately 8.82% or 6.0 million people had diabetes, 1.64% or 1.1 million had unscreened diabetes, 2.74% or 1.9 million had undiagnosed diabetes, and 0.85% or 0.6 million had uncontrolled diabetes (S2 Table).

**Table 1. Demographic characteristics of analytic sample and prevalence of diabetes, NHES-V Thailand, 2014.**

| | N | Percent | Diabetes | |
| | | | Percent | SE |
|---|---|---|---|---|
| Age Standardized | | | 8.82 | 0.31 |
| Crude | | | 11.1 | 0.34 |
| **Age (years)** | | | | |
| 20–29 | 1125 | 15.8 | 2.86 | 0.6 |
| 30–39 | 1751 | 17.4 | 4.71 | 0.61 |
| 40–49 | 3041 | 22.4 | 9.13 | 0.74 |
| 50–59 | 3418 | 24.0 | 15.5 | 0.83 |
| 60–69 | 3756 | 11.5 | 20.8 | 0.89 |
| 70+ | 2572 | 8.8 | 18.9 | 1.12 |
| **Sex** | | | | |
| Female | 9102 | 52.4 | 9.26 | 0.43 |
| Male | 6561 | 47.6 | 8.28 | 0.45 |
| **BMI (kg/m2)** | | | | |
| Underweight | 1051 | 6.8 | 6.69 | 1.15 |
| Normal | 8135 | 53.4 | 8.91 | 0.44 |
| Overweight | 4863 | 28.7 | 14.5 | 0.68 |
| Obese | 1614 | 11 | 15.6 | 1.15 |
| **Religion** | | | | |
| Buddhist | 14649 | 94 | 8.96 | 0.33 |
| Not Buddhist | 1014 | 6 | 6.85 | 0.89 |
| **Highest Educational Level** | | | | |
| Primary or less | 10293 | 57.7 | 10.0 | 0.77 |
| Low secondary | 1448 | 12.2 | 9.38 | 0.96 |
| High secondary or vocational | 2444 | 19.3 | 7.47 | 0.63 |
| University | 1478 | 10.8 | 7.03 | 0.92 |
| **Geography** | | | | |
| Rural | 7416 | 56 | 9.06 | 0.47 |
| Urban | 8247 | 44 | 8.63 | 0.4 |
| **Region** | | | | |
| Bangkok | 3423 | 17.3 | 8.07 | 0.73 |
| South | 3753 | 27 | 6.11 | 0.5 |
| North | 3601 | 29.1 | 7.52 | 0.63 |
| Central | 2658 | 12.7 | 10.8 | 0.71 |
| Northeast | 2228 | 13.8 | 9.53 | 0.66 |
| Sample size | 15663 | | 2255 | |

SE = standard error. BMI = body mass index. Sample weights were incorporated to adjust the percentage estimates in NHES-V sample for unequal probabilities of selection. BMI categories were: underweight (BMI < 18.5 kg/m^2), normal (18.5 ≤ BMI < 25), overweight (25 ≤ BMI < 30), and obese (BMI ≤ 30). Estimates for overall population and by sex, BMI, religion, educational level, geography, and region were age-standardized using five-year categories between 20–70+ using the 2010 Thai Census population estimates.

The diabetes care cascade, stratified by region, is shown in Fig 3. There is regional variation in cascade attrition. The largest gaps at screening and diagnosis occurred in Central (screening 38.7% [95% CI 30.4% to 47.0%], diagnosis 32.7% [95% CI 29.1% to 36.3%]) and Northeast regions (screening 28.9% [95% CI 21.1% to 36.7%], diagnosis 38.7% [95% CI 32.6% to 44.8%]), and the smallest occurred in South region (screening 16.9% [91% CI 5.0% to 28.8%], diagnosis

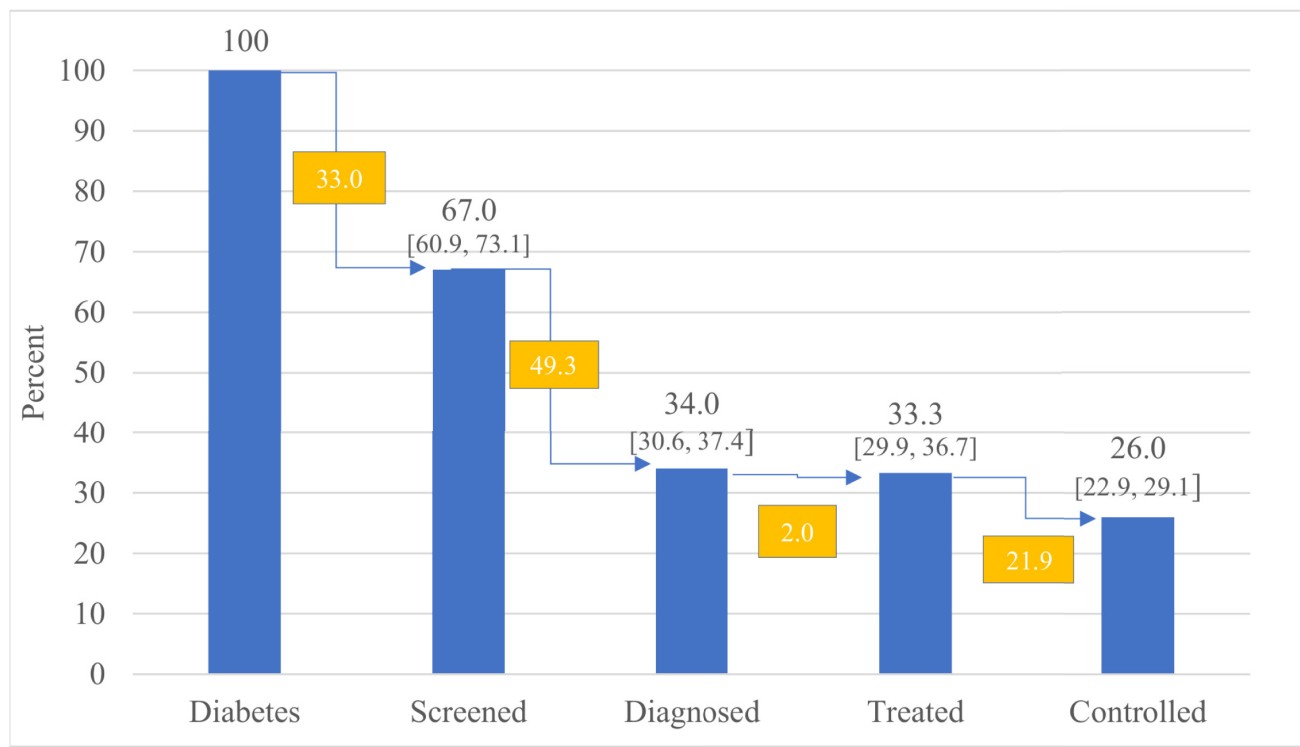

**Fig 2. Diabetes care cascade, Thailand 2014.** Point estimates are shown, with 95% confidence intervals in brackets. Among all people with diabetes, 67.0% were ever screened for diabetes (33.0% relative loss), 34.0% were ever diagnosed (49.3% loss), 33.3% were ever treated (2.0% loss), and 26.0% were controlled with fasting plasma glucose <183 mg/dL (21.9% relative loss). Unmet need was 74.0% across the care cascade.

31.5% [95% CI 18.5% to 44.5%]). Unmet need ranged from 58.4% (95% CI 45.0% to 71.8%) in South to 78.0% (95% CI 73.0% to 83.0%) in Northeast region.

## Logistic regression to explore care cascade attrition

We created regression models across the care continuum to understand if care cascade attrition was explained by individual level variables, or by health system level variables (Table 2). Each ten-year increase in age was associated with a higher likelihood of being screened (OR 2.62, 95% CI 2.12 to 3.25), diagnosed (OR 1.76, 95% CI 1.56 to 1.98), and controlled (OR 1.80, 95% CI 1.61 to 2.01). Male sex was associated with decreased likelihood of all outcomes screened (OR 0.38, 95% CI 0.23 to 0.61) and diagnosed (OR 0.65, 95% CI 0.50 to 0.86), but not statistically significantly associated with controlled. There was a trend towards increased likelihood of screened, diagnosed, and controlled for increasing BMI, which was most pronounced for diagnosed and controlled. Lastly, two health system factors proved important related to outcomes, with variation of availability by region (S2 Fig). Increased density of medical staff was associated with higher likelihood of screened (OR 2.49, 95% CI 1.03 to 6.01), diagnosed (OR 2.40, 95% CI 1.41 to 4.08) and controlled (OR 1.82, 95% CI 1.10 to 2.99), and increased density of health centers, but not hospitals, was associated with higher likelihood of screened (OR 2.33, 95% CI 1.24 to 4.39), diagnosed (OR 1.58, 95% CI 1.12 to 2.24), and controlled (OR 1.56, 95% CI 1.13 to 2.15).

To examine which independent variables have statistically different coefficients across care cascade outcomes, we used the Brant test. Northeast region, age, sex, BMI, and density of

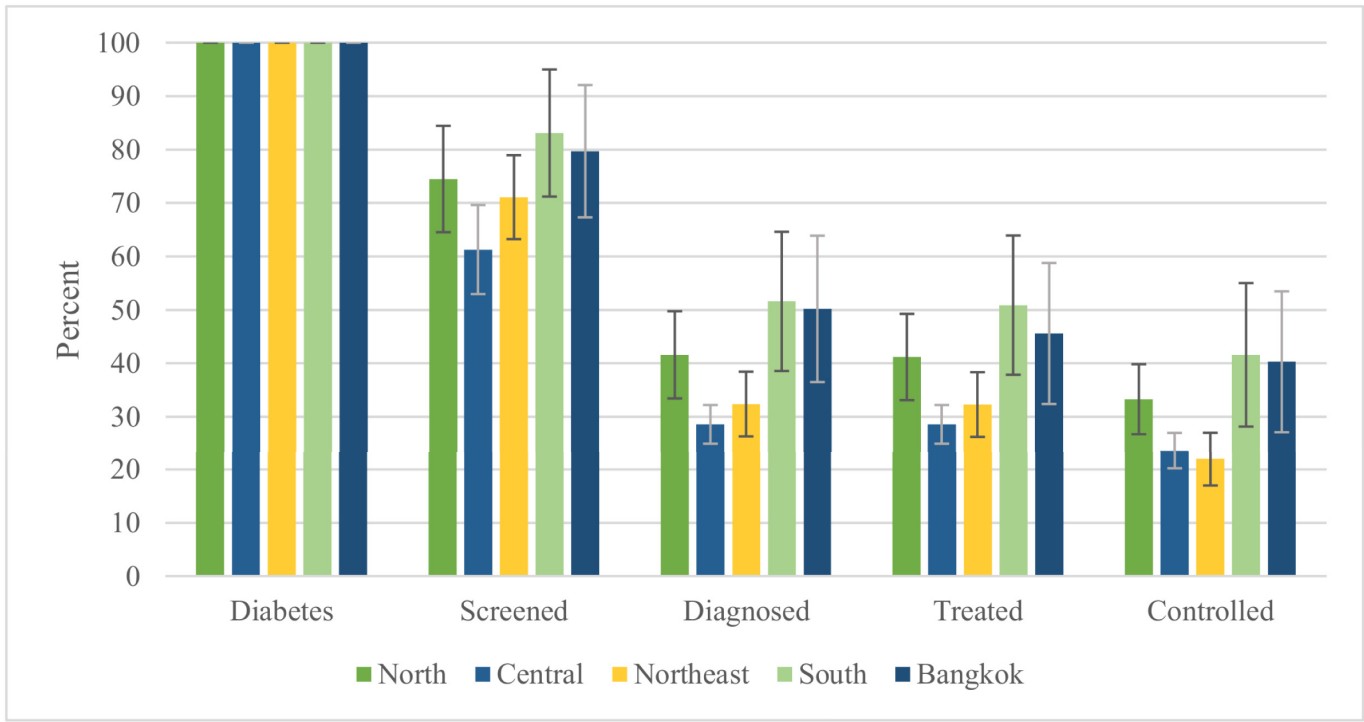

**Fig 3. Regional diabetes care cascade, Thailand 2014.** Point estimates are shown, with 95% confidence interval bars. Within different regions (North, Central, Northeast, South, Bangkok), people with diabetes had different rates of attrition across the care cascade. Among people with diabetes, the Northeast had the lowest rates of control (21.8%), while South had the highest rates of control (47.9%).

hospitals, staff, and health centers had statistically different coefficients across screened, diagnosed, and treated (S4 Table).

In a sensitivity analysis with nested logistic regression, the outcome of diagnosed was similarly associated with age (OR 1.50, 95% CI 1.30 to 1.72), rising BMI, and density of medical staff (OR 2.17, 95% CI 1.24 to 3.81). For controlled, age (OR 1.59, 95% CI 1.32 to 1.91) remained significant, but health system factors did not.

## Discussion

This study had several key findings. First, we identified significant unmet need for diabetes care in the Thai adult population, with 74% of those with diabetes having an unmet need for care across levels of screening, diagnosis, treatment, or control. Second, the high unmet need for diabetes care was found to be largely attributable to loss at the stages of screening and diagnosis, which each contributed 33% to total unmet need. Third, although differences were not statistically significant, we found some suggestive evidence of regional variation in cascade performance, with unmet need ranging from 58.4% in South to 78.0% in the Northeast region. Fourth, across the sequential care cascade outcomes, we found that variation in cascade performance was explained both by demographic and health systems factors.

Our absolute losses of -33% at screening, and -66% at diagnosis in Thailand are slightly better for screening and slightly worse for diagnosis compared to other diabetes care cascade studies in South Africa (absolute losses -45% at screening, -60% at diagnosis), and globally in 28 low-and-middle income countries (absolute losses -37% at screening, -56% at diagnosis)

**Table 2. Factors associated with diabetes care cascade retention, Thailand 2014.** Un-nested logistic regression.

| | Screened | | | Diagnosed | | | Controlled | | |
|---|---|---|---|---|---|---|---|---|---|
| | aOR | 95% CI | p value | aOR | 95% CI | p value | aOR | 95% CI | p value |
| **Region** | | | | | | | | | |
| Northeast | 1 | | | 1 | | | 1 | | |
| Bangkok + Central | 0.60 | 0.30 1.20 | 0.15 | 0.84 | 0.56 1.27 | 0.42 | 1.18 | 0.80 1.75 | 0.40 |
| South | 0.83 | 0.36 1.93 | 0.67 | 1.23 | 0.74 2.02 | 0.43 | 1.35 | 0.84 2.18 | 0.21 |
| North | 0.74 | 0.38 1.44 | 0.37 | 1.01 | 0.68 1.5 | 0.97 | 1.36 | 0.92 2.00 | 0.12 |
| **Age** | | | | | | | | | |
| Age in 10 year increments | 2.62 | 2.12 3.25 | <0.001 | 1.76 | 1.56 1.98 | <0.001 | 1.80 | 1.61 2.01 | <0.001 |
| **Sex** | | | | | | | | | |
| Female | 1 | | | 1 | | | 1 | | |
| Male | 0.38 | 0.23 0.61 | <0.001 | 0.65 | 0.5 0.86 | 0.002 | 0.81 | 0.63 1.05 | 0.12 |
| **BMI** | | | | | | | | | |
| Underweight | 0.46 | 0.17 1.21 | 0.12 | 0.53 | 0.24 1.17 | 0.12 | 0.53 | 0.24 1.14 | 0.10 |
| Normal | 1 | | | 1 | | | 1 | | |
| Overweight | 2.35 | 1.39 3.98 | 0.001 | 1.58 | 1.17 2.13 | 0.003 | 1.47 | 1.11 1.94 | 0.007 |
| Obese | 1.28 | 0.68 2.43 | 0.44 | 1.62 | 1.11 2.37 | 0.01 | 1.75 | 1.22 2.51 | 0.002 |
| **Highest Educational Level** | | | | | | | | | |
| Primary or Lower | 1 | | | 1 | | | 1 | | |
| Low Secondary | 1.48 | 0.67 3.29 | 0.33 | 0.71 | 0.43 1.15 | 0.16 | 0.84 | 0.51 1.38 | 0.49 |
| High Secondary or Vocational | 1.95 | 0.94 4.05 | 0.07 | 0.99 | 0.63 1.55 | 0.96 | 1.10 | 0.71 1.69 | 0.67 |
| University | 1.03 | 0.42 2.54 | 0.94 | 0.76 | 0.42 1.37 | 0.36 | 0.87 | 0.47 1.62 | 0.66 |
| **Geography** | | | | | | | | | |
| Rural | 1 | | | 1 | | | 1 | | |
| Urban | 0.92 | 0.56 1.51 | 0.74 | 0.90 | 0.68 1.19 | 0.46 | 0.85 | 0.64 1.12 | 0.25 |
| **Health System** | | | | | | | | | |
| Hospital per Population, standardized | 0.63 | 0.39 1.51 | 0.74 | 0.76 | 0.58 1.00 | 0.05 | 0.82 | 0.64 1.06 | 0.14 |
| Health Center per Population, Standardized | 2.33 | 1.24 4.39 | 0.01 | 1.58 | 1.12 2.24 | 0.01 | 1.56 | 1.13 2.15 | 0.01 |
| Staff per Population, standardized | 2.49 | 1.03 6.01 | 0.04 | 2.40 | 1.41 4.08 | 0.001 | 1.82 | 1.10 2.99 | 0.02 |
| Public Health Nurses per Population, Standardized | 0.94 | 0.53 1.67 | 0.84 | 0.71 | 0.50 1.00 | 0.05 | 0.79 | 0.57 1.09 | 0.15 |
| Subpopulation (n) | 2255 | | | 2255 | | | 2255 | | |

Multivariable adjusted odds ratios estimated using logistic regression with un-nested denominators at each stage. Analysis incorporated sample weights.

aOR = adjusted odds ratio. BMI = body mass index. BMI categories were: underweight (BMI < 18.5 kg/m^2), normal ($18.5 \leq BMI < 25$), overweight ($25 \leq BMI < 30$), and obese ($BMI \leq 30$). For health system variables (population per hospital, population per staff, population per health center, population per public health nurses), values were standardized so a one unit increase represents a one standard deviation increase from the mean.

[12–14]. The United States fares the worst among these cascades, with an absolute loss of -72% at diagnosis [16].

Earlier studies presented very low levels of unmet health care need in Thailand, at less than two percent of the population, based on individual subjective assessment of personal illness and utilization need [19,20]. Given our 74.0% unmet need for only diabetes, we argue actual unmet health care need is much larger than previously reported, and requires objective assessment to complement subjective reports. Multiple factors contribute to Thailand's loss at screening and diagnosis. While Thailand has implemented universal coverage of health insurance since 2002 which reduced patient financial burdens and increased healthcare access, concerns remain around long wait times and low service quality in primary care settings, which may deter some patients from accessing screening and diagnosis [21]. Furthermore, early

stages of diabetes can be asymptomatic, so that even if a patient attends a clinic visit, the physician must have a higher degree of suspicion to screen and diagnose diabetes, compared to symptomatic conditions that patients will mention themselves [22].

Regional variation in cascade progression was not significant in multivariable models, suggesting the differences may be due to a combination of demographic and health system factors. This has also been suggested in other studies examining geographic differences in health outcomes in Thailand, after implementation of universal health insurance. While overall mortality has steadily declined since 2002, the faster rate of decline in Bangkok compared to the North and Northeast regions has been surmised to be related to the higher poverty and lower health workforce density in the latter two regions [23]. This is consistent with our results, which showed that higher health staff density was associated with a higher OR of progressing through the cascade to diagnosis and control. Regional differences in the proportion of people on the Civil Servant Medical Benefit Scheme or Social Security Scheme (government employees and private sector, relatively high income) vs the Universal Coverage Scheme (informal employment sector, relatively low income), may also influence cascade progression as healthcare utilization and some medication access has shown to differ among the three insurance schemes [24,25]. Due to small sample size, we were not able to examine interactions between region and health system factors. Additional studies are needed to better understand the extent to which regional variation in cascade performance in Thailand may be driven by regional differences in health system characteristics.

While care cascades are a useful way to measure quality and monitor progress at the health system level, there are many other socioeconomic, interpersonal, and structural factors in LMIC which influence good outcomes for diabetes and are not adequately captured, as conceptualized in the socio-ecologic model for health [26]. Political instability, lack of public infrastructure such as roads, cultural norms around food, barriers to meaningful physical activity, competing demands for limited resources at the individual level, and personal conceptual models of illness may all influence if a person develops diabetes, and how far through the cascade they progress. For example, in Thailand, one qualitative study explored how diabetes was viewed as a natural part of aging in the Buddhist life-course, which may impede treatment uptake [27]. Successful interventions will account for this complexity.

Our study highlights the need for stronger investment to strengthen primary health care in Thailand. An independent assessment after a decade of the Thai Universal Coverage Scheme (UCS) indicated that the focus on curative care may have contributed to lower resources for public health functions [28]. While several national policies to improve diabetes screening and care have been passed, and a dedicated "chronic care fund" was established under UCS to strengthen screening and primary care for diabetes and hypertension in 2011, large gaps remain in disease detection. Future steps might include expanding primary health care clinics and staff, in addition to auxiliary health providers like community pharmacists, who in prior studies have successfully managed diabetes and hypertension in conjunction with primary care providers. [21,29]. Better health information systems that allow every Thai to access their personal health information, including diabetes risk and screening records, could also contribute to reducing unmet need.

This study had several limitations. First, we were unable to distinguish between type I and type II diabetes mellitus—however these conditions are not routinely disaggregated in other nationwide studies as the cascade targets are similar [16]. Additionally, in adult populations the overwhelming majority of people with diabetes are type II. Second, the single measurement of fasting plasma glucose may either overestimate the prevalence of diabetes if participants were not truly fasting, or underestimate it compared to an oral glucose tolerance test. A prior study in Thailand comparing fasting plasma glucose and the oral glucose tolerance test showed

that fasting plasma glucose missed up to 46.3% of all prediabetes and 4.7% of all diabetes [30]. Third, participants with diabetes on treatment may be significantly more likely to report past diabetes screening or diagnosis, compared to participants with diabetes not on treatment. This would skew attrition to occur earlier (screening/diagnosis) rather than later (treatment/control). Fourth, given the cross-sectional study design, we were not able to examine the association of attrition across stages of the cascade with health outcomes or assess the temporal ordering of cascade steps. Therefore, it is possible that for some individuals, screening occurred prior to the development of diabetes, leading to an overestimate of attrition between the screening and diagnosis steps of the cascade. Future research should evaluate how unmet need for diabetes care affects progression to diabetes complications and associated health care costs through a prospective cohort.

In this nationally representative study, diabetes and prediabetes affected one in four adults over the age of 20 in Thailand. The care cascade is a helpful framework to understand where people with diabetes are lost in the healthcare system, with the largest drop offs at screening and diagnosis. Even with universal health insurance coverage, unmet need remained. Achieving screened, diagnosed, and controlled diabetes was more likely in older people, and in areas with increased density of medical staff or health centers. Future interventions should target increased screening and diagnosis of diabetes in Thailand.

## Supporting information

**S1 Fig. Flow chart of study participants, NHES-V Thailand, 2014.**
(EPS)

**S2 Fig. Number of health facilities and staff by region, Thailand 2014.**
(EPS)

**S1 Table. Prevalence of normoglycemia, prediabetes, and diabetes in Thailand, 2014.**
SE = standard error. BMI = body mass index. Normoglycemia = fasting plasma glucose < 100 mg/dL and not on treatment. Prediabetes = fasting plasma glucose ≥ 100 mg/dL and not on treatment. BMI categories were: underweight (BMI < 18·5 kg/m^2), normal (18·5 ≤ BMI < 25), overweight (25 ≤ BMI < 30), and obese (BMI ≤ 30). Estimates for overall population and by sex, BMI, religion, educational level, geography, and region were age-standardized using five-year categories between 20–70+ using the 2010 Thai Census population estimates.
(DOCX)

**S2 Table. Prevalence of unscreened, undiagnosed, untreated, and uncontrolled diabetes, among total Thai population 2014.** SE = standard error. BMI = body mass index. BMI categories were: underweight (BMI < 18·5 kg/m^2), normal (18·5 ≤ BMI < 25), overweight (25 ≤ BMI < 30), and obese (BMI ≤ 30). Estimates for overall population and by sex, BMI, religion, educational level, geography, and region were age-standardized using five-year categories between 20–70+ using the 2010 Thai Census population estimates. Source: NHES-V
(DOCX)

**S3 Table. Factors associated with diabetes care cascade retention, Thailand 2014. Nested logistic regression.** Multivariable adjusted odds ratios estimated using continuation ratio logit model with coefficients freely varying across stages. Analysis incorporated sample weights. aOR = adjusted odds ratio. BMI = body mass index. BMI categories were: underweight (BMI < 18·5 kg/m^2), normal (18·5 ≤ BMI < 25), overweight (25 ≤ BMI < 30), and obese (BMI ≤ 30). For health system variables (hospitalization per population, staff per population, health center per population, public health nurses per population), values were standardized

so a one unit increase represents a one standard deviation increase from the mean. Source: NHES-V.
(DOCX)

**S4 Table. Brant test on independent variable coefficients across outcomes of screened, diagnosed, and controlled.** Brant test of parallel regression assumption tests whether coefficients for a specific independent variable is statistically different across sequential ordinal outcomes, in this case screened, diagnosed, or controlled diabetes. The null hypothesis is that all coefficients are the same. P values > 0·05 indicate the null hypothesis is true, ie coefficients are the same across outcomes. P values <0·05 indicate evidence to reject the null hypothesis, ie coefficients are different across outcomes.
(DOCX)

**S5 Table. STROBE checklist.** STROBE checklist for cross-sectional studies.
(DOC)

## Acknowledgments

We would like to thank Katelyn M. Berry, Adna Glusac, and Michelle Shu for their input on an earlier version of this manuscript, Supipa Buranasiri for her assistance in translating the NHES-V survey questionnaire to English, and Wasin Laohavinij for his assistance with the literature review. Lastly, we would like to thank the participants of the NHES-V survey.

## Author Contributions

**Conceptualization:** Lily D. Yan, Piya Hanvoravongchai, Andrew C. Stokes.

**Data curation:** Lily D. Yan, Wichai Aekplakorn, Suwat Chariyalertsak, Pattapong Kessomboon, Sawitri Assanangkornchai, Surasak Taneepanichskul, Nareemarn Neelapaichit.

**Formal analysis:** Lily D. Yan, Wichai Aekplakorn, Andrew C. Stokes.

**Investigation:** Lily D. Yan, Andrew C. Stokes.

**Methodology:** Piya Hanvoravongchai, Andrew C. Stokes.

**Project administration:** Lily D. Yan.

**Software:** Lily D. Yan.

**Supervision:** Piya Hanvoravongchai, Andrew C. Stokes.

**Validation:** Lily D. Yan, Andrew C. Stokes.

**Visualization:** Lily D. Yan.

**Writing – original draft:** Lily D. Yan.

**Writing – review & editing:** Lily D. Yan, Piya Hanvoravongchai, Wichai Aekplakorn, Suwat Chariyalertsak, Pattapong Kessomboon, Sawitri Assanangkornchai, Surasak Taneepanichskul, Nareemarn Neelapaichit, Andrew C. Stokes.

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
