## [Decision Letter · Decision Letter 0]

24 Sep 2019

PONE-D-19-22038

Universal coverage but unmet need: national and regional estimates of attrition across the diabetes care continuum in Thailand

PLOS ONE

Dear Dr. Stokes,

Thank you for submitting your manuscript to PLOS ONE. After careful consideration, we feel that it has merit but does not fully meet PLOS ONE’s publication criteria as it currently stands. Therefore, we invite you to submit a revised version of the manuscript that addresses the points raised during the review process.

Please carefully consider each of the concerns raised by the reviewers in revising the manuscript.

We would appreciate receiving your revised manuscript by Nov 08 2019 11:59PM. To enhance the reproducibility of your results, we recommend that if applicable you deposit your laboratory protocols in protocols.io, where a protocol can be assigned its own identifier (DOI) such that it can be cited independently in the future. For instructions see: http://journals.plos.org/plosone/s/submission-guidelines#loc-laboratory-protocols

We look forward to receiving your revised manuscript.

Kind regards,

Nayu Ikeda, Ph.D.

Academic Editor

PLOS ONE

Journal Requirements:

1. We note that you have stated that you will provide repository information for your data at acceptance. Should your manuscript be accepted for publication, we will hold it until you provide the relevant accession numbers or DOIs necessary to access your data. If you wish to make changes to your Data Availability statement, please describe these changes in your cover letter and we will update your Data Availability statement to reflect the information you provide.

Reviewers' comments:

Reviewer's Responses to Questions

**Comments to the Author**

1. Is the manuscript technically sound, and do the data support the conclusions?

Reviewer #1: No

Reviewer #2: Yes

2. Has the statistical analysis been performed appropriately and rigorously? 

Reviewer #1: No

Reviewer #2: Yes

3. Have the authors made all data underlying the findings in their manuscript fully available?

Reviewer #1: No

Reviewer #2: No

4. Is the manuscript presented in an intelligible fashion and written in standard English?

Reviewer #1: Yes

Reviewer #2: Yes

5. Review Comments to the Author

Reviewer #1: The research question and the idea of the paper are interesting and relevant. However, the statistical implementation and interpretation of empirical results are erroneous for multiple reasons:

1) The diabetes care cascades are missing confidence intervals or any measure of precision. Therefore, no conclusions about differences across cascade stages should be made.

2) Point 1 also applies to regional differences at different cascade stages. Point estimates seem to suggest differences across regions, but the reader remains clueless whether the differences are statistically meaningful.

3) From the text and the table, it is not exactly clear how the estimation specification was set-up. For example, there is no mention that regional fixed effects were included. However, in the conclusion the authors mention that regions were not significant in the regression analysis. While this point regards transparency it also withholds the reader relevant information - the coefficients on regions are of interest and should be included in table 2. Further, the choice of using a continuous 10-year age increment variable rather than 10-year age bracket fixed effects is not comprehensible.

4) These first 4 points reduce the credibility of the claims made in the first discussion paragraph.

5) Confidence intervals and/or p-values are not included in the text.

6) The study mentions some "preliminary analysis of diabetes", however, it remains unknown what this analysis entails.

In Addition, there are some conceptual aspects which could improve the analysis - where the first point is of much greater relevance than the second one.

1) The most interesting aspect of the paper is the regional variation in health system performance. However, the authors do not at all tease out this point to the extent possible. For example, the inclusion of interactions of health system factors and regional indicators in the regression model, would create much more detailed insights into relevant health system factors across regions to explain the considered losses. While this may look messy in a regression output table, such comparisons may be nicely visualized.

2) The NHES-V is a repeated survey. In order to explore the impact of universal health coverage, the authors might want to look at the evolution of care cascades over time.

Reviewer #2: The authors worked on an interesting topic and on a large database. The paper is well structured. The methodology is clear. The results are presented with interesting details. However, they could improve the quality of their paper.

Comments

--Lines 218-223 : Harmonize the numbers on Chart A and in the legend

--Line 217 : Supplementary Table 2 : I suggest to the authors to present the cascade levels : screened, diagnosed, treated and controlled, instead of the opposite. It will facilitate the analysis of the other results, especially the table 2.

I also suggest them to present p-value in table 2 to better describe the raw relationship between variables and cascade levels. This will allow a better understanding the discussion on multivariate analysis "Regional variation in cascade progression was not significant after multivariable adjustment... ".

--Lines 333-335 "Second, the single measurement of fasting plasma glucose may not have captured all people with diabetes, and underestimate the prevalence of diabetes…all diabetes.

The authors should qualify their assertions. Failure to perform an oral tolerance glucose test for prediabetics may underestimate the frequency of diabetes as they mentioned. Taking into account of a single measure could rather overestimate this frequency. Some participants could have not respected fasting for the first measure.

---The authors showed that the maximum attrition was on the diagnosis level. In the discussion (line 335-340), they could more discuss clearly this result because the model of the cascade has some limits. The status of the participant may have changed between the last screening and the date of the diagnosis performed by the study. Being screened, but undiagnosed may not be only linked to the health system weakness.

---There are some points in the cascade the authors could compare with results from other regions. Almost all people diagnosed had been ever treated and nearly three-quarters of those treated had a fasting plasma glucose level of less than 1.83 g / l. These results seem better than others.

6. PLOS authors have the option to publish the peer review history of their article (what does this mean?). If published, this will include your full peer review and any attached files.

Reviewer #1: No

Reviewer #2: No

---

## [Author Response · Author response to Decision Letter 0]

25 Oct 2019

Reviewer Comments to the Author

Reviewer #1: The research question and the idea of the paper are interesting and relevant. However, the statistical implementation and interpretation of empirical results are erroneous for multiple reasons:

1) The diabetes care cascades are missing confidence intervals or any measure of precision. Therefore, no conclusions about differences across cascade stages should be made.

a. RESPONSE: Thank you for your suggestion. We have modified Fig2 to include not only the point estimate, but also the 95% confidence interval for each stage. We have also edited the text in the Results section to reflect this.

b. Lines 216-219: Figure 2 shows the diabetes care cascade. Among all people with diabetes, 67.0% (95% CI 60.9% to 73.1%) were screened, 34.0% (95% CI 30.6% to 37.4%) were diagnosed, 33.3% (95% CI 29.9% to 36.7%) were treated, and 26.0% (95% CI 22.9% to 29.1%) were controlled. The unmet need for diabetes was 74.0% (95% CI 70.9% to 77.1%).

2) Point 1 also applies to regional differences at different cascade stages. Point estimates seem to suggest differences across regions, but the reader remains clueless whether the differences are statistically meaningful. 

a. RESPONSE: Thank you. We have modified Fig3 to also include the 95% confidence intervals as error bars, and edited the associated text.

b. Lines 232-238: The diabetes care cascade, stratified by region, is shown in Figure 3. There is regional variation in cascade attrition. The largest gaps at screening and diagnosis occurred in Central (screening 38.7% [95% CI 30.4% to 47.0%], diagnosis 32.7% [95% CI 29.1% to 36.3%]) and Northeast regions (screening 28.9% [95% CI 21.1% to 36.7%], diagnosis 38.7% [95% CI 32.6% to 44.8%]), and the smallest occurred in South region (screening 16.9% [91% CI 5.0% to 28.8%], diagnosis 31.5% [95% CI 18.5% to 44.5%]). Unmet need ranged from 58.4% (95% CI 45.0% to 71.8%) in South to 78.0% (95% CI 73.0% to 83.0%) in Northeast region.

3) From the text and the table, it is not exactly clear how the estimation specification was set-up. For example, there is no mention that regional fixed effects were included. However, in the conclusion the authors mention that regions were not significant in the regression analysis. While this point regards transparency it also withholds the reader relevant information - the coefficients on regions are of interest and should be included in table 2. 

a. RESPONSE: Thank you. In the Methods text, although we listed other independent variables, we did neglect to explicitly list “region” and “education”. We have corrected this (see “b” below). The reviewer mentions that the coefficients on regions are not included in Table 2. However, in Table 2 we did include all of the independent variables (line 254). Perhaps some of the confusion comes from the fact that we combined Bangkok (the city) and Central (the region). This combination was suggested specifically by Dr Wichai Aekplakorn, who is one of the principal researchers on NHES and very familiar with the survey design and rollout. Given the location of Bangkok within Central, and small sample size with Bangkok, we have combined the two. We have added a section to the Methods text to explicitly mention this (see “c” below). 

b. Lines 155-162: The main outcomes of interest in this study were people with diabetes who were screened, diagnosed, treated, and controlled. Independent variables included region, individual factors (age, sex, BMI, highest educational level) and health system factors (healthcare provider density, hospital density, health center density). Information on family history and treatment adherence could not be included due to the amount of missing data. 

c. Lines 158-160: For region, we combined data from Bangkok and Central, given the city is nested within the region and the relatively small sample size.

4) Further, the choice of using a continuous 10-year age increment variable rather than 10-year age bracket fixed effects is not comprehensible. 

a. RESPONSE: We apologize for the confusion. We have simply rescaled the continuous age variable so that the adjusted odds ratio in the model corresponds to a 10-unit increase in age as opposed to a 1-unit increase in age. Other than affecting the interpretation of the beta coefficient, the rescaling does not affect the model results. We have adjusted the text to clarify this point. 

b. Line 160: Age was included as a continuous variable in 10 year increments.

5) These first 4 points reduce the credibility of the claims made in the first discussion paragraph.

a. RESPONSE: We hope that addressing the concerns above will improve the interpretability of our Results, and the credibility of our Discussion. We have tempered the language about the regional trends, as not all the differences were statistically different.

b. Lines 282-285: Third, although differences were not statistically significant, we found some suggestive evidence of regional variation in cascade performance, with unmet need ranging from 58.4% in South to 78.0% in the Northeast region.

6) Confidence intervals and/or p-values are not included in the text.

a. RESPONSE: Thank you. Please see above for modifications to include confidence intervals in the text for the care cascades. For the regressions, to avoid duplication of text and table, we did not originally include confidence intervals. However, per your suggestion, we have now included all 95% CI for reported odds ratios.

b. Examples, lines 247 to 250: Each ten-year increase in age was associated with a higher likelihood of being screened (OR 2.62, 95% CI 2.12 to 3.25), diagnosed (OR 1.76, 95% CI 1.56 to 1.98), and controlled (OR 1.80, 95% CI 1.61 to 2.01).

7) The study mentions some "preliminary analysis of diabetes", however, it remains unknown what this analysis entails.

a. RESPONSE: Thank you for your point. The preliminary analysis was the care cascade presented in Figure 2. We have edited the text in the Methods section to clarify this.

b. Line 174-176: Based on initial analysis of the diabetes care cascade in the present study, we modeled the following outcomes: 1) probability of screening conditional on diabetes; 2) probability of diagnosis conditional on diabetes; 3) probability of control conditional on diabetes.

c. Line 177-178: We did not separately model treatment as the care cascade preliminary analysis revealed nearly complete progression of the sample between diagnosis and treatment.

In Addition, there are some conceptual aspects which could improve the analysis - where the first point is of much greater relevance than the second one.

1) The most interesting aspect of the paper is the regional variation in health system performance. However, the authors do not at all tease out this point to the extent possible. For example, the inclusion of interactions of health system factors and regional indicators in the regression model, would create much more detailed insights into relevant health system factors across regions to explain the considered losses. While this may look messy in a regression output table, such comparisons may be nicely visualized.

a. RESPONSE: Thank you for your suggestion. Including an interaction of health system factors (4) with regions (4 with Bangkok+Central combined) would result in 16 interaction terms. Given our sample size of 2255 people with diabetes in the NHES survey, many of these cells would have too few people for the coefficients to be meaningful. However, we have incorporated a plot of the number of hospitals, stand alone health clinics, doctors, nurses, and public health nurses by region in a new Supplemental Figure 2 and include reference to it in the text.

b. Lines 254-255: Lastly, two health system factors proved important related to outcomes, with variation of availability by region (Supplemental Figure 2).

c. Lines 317-321: Due to small sample size, we were not able to examine interactions between region and health system factors. Additional studies are needed to better understand the extent to which regional variation in cascade performance in Thailand may be driven by regional differences in health system characteristics.

2) The NHES-V is a repeated survey. In order to explore the impact of universal health coverage, the authors might want to look at the evolution of care cascades over time.

a. RESPONSE: Thank you for your suggestion. We agree that longitudinal analysis would better capture the effect of universal health coverage, and consider it a promising avenue for future research. 

Reviewer #2: The authors worked on an interesting topic and on a large database. The paper is well structured. The methodology is clear. The results are presented with interesting details. However, they could improve the quality of their paper.

Comments

1) Lines 218-223 : Harmonize the numbers on Chart A and in the legend

a. RESPONSE: Thank you for pointing this out. We have corrected the outdated legend numbers to match the Figure 2 values.

b. Lines 227-230: Point estimates are shown, with 95% confidence intervals in brackets. Among all people with diabetes, 67.03% were ever screened for diabetes (33.02.7% relative loss), 34.06% were ever diagnosed (49.38.6% loss), 33.39% were ever treated (2.0% loss), and 26.03% were controlled with fasting plasma glucose <183 mg/dL (21.92.4% relative loss). Unmet need was 74.03.7% across the care cascade.

2) Line 217 : Supplementary Table 2 : I suggest to the authors to present the cascade levels : screened, diagnosed, treated and controlled, instead of the opposite. It will facilitate the analysis of the other results, especially the table 2.

a. RESPONSE: Thank you for your suggestion. The rationale behind choosing to display unscreened, undiagnosed, untreated, uncontrolled, and controlled in Supplementary Table 2 is to show the mutually exclusive and exhaustive cascade categories of diabetes. The absolute percentages shown in the age-standardized line add up to the overall prevalence of diabetes in Thailand (8.8%). This also focuses on unmet need, and complements the main presentation of the data.

3) I also suggest them to present p-value in table 2 to better describe the raw relationship between variables and cascade levels. This will allow a better understanding the discussion on multivariate analysis "Regional variation in cascade progression was not significant after multivariable adjustment... ".

a. RESPONSE: Thank you for your suggestion. Table 2 (line 259) does have the p values displayed, in the column labeled “p”. As this did not seem to be clear, we have revised the column title from “p” to “p-value”.

b. Line 261 Table 2

4) Lines 333-335 "Second, the single measurement of fasting plasma glucose may not have captured all people with diabetes, and underestimate the prevalence of diabetes…all diabetes.” The authors should qualify their assertions. Failure to perform an oral tolerance glucose test for prediabetics may underestimate the frequency of diabetes as they mentioned. Taking into account of a single measure could rather overestimate this frequency. Some participants could have not respected fasting for the first measure.

a. RESPONSE: Thank you, we have incorporated your suggestion.

b. Line 347-350: Second, the single measurement of fasting plasma glucose may either overestimate the prevalence of diabetes if participants were not truly fasting, or underestimate it compared to an oral glucose tolerance test.

5) The authors showed that the maximum attrition was on the diagnosis level. In the discussion (line 335-340), they could more discuss clearly this result because the model of the cascade has some limits. The status of the participant may have changed between the last screening and the date of the diagnosis performed by the study. Being screened, but undiagnosed may not be only linked to the health system weakness.

a. RESPONSE: We agree. A patient who has been screened in the past (years ago) may have developed diabetes between then and the date of the survey, without intervening screening in between. We have attempted to describe this limitation more clearly. 

b. Line 355-360: Fourth, given the cross-sectional study design, we were not able to examine the association of attrition across stages of the cascade with health outcomes or assess the temporal ordering of cascade steps. Therefore, it is possible that for some individuals, screening occurred prior to the development of diabetes, leading to an overestimate of attrition between the screening and diagnosis steps of the cascade. 

6) There are some points in the cascade the authors could compare with results from other regions. Almost all people diagnosed had been ever treated and nearly three-quarters of those treated had a fasting plasma glucose level of less than 1.83 g / l. These results seem better than others.

a. RESPONSE: We have compared our care cascade results with other diabetes care cascades in South Africa, 28 other low and middle income countries, and the United States.

b. Lines 287-292: Our absolute losses of -33% at screening, and -66% at diagnosis in Thailand are slightly better for screening and slightly worse for diagnosis compared to other diabetes care cascade studies in South Africa (absolute losses -45% at screening, -60% at diagnosis), and globally in 28 low-and-middle income countries (absolute losses -37% at screening, -56% at diagnosis) [12–14]. The United States fares the worst among these cascades, with an absolute loss of -72% at diagnosis [16].

---

## [Decision Letter · Decision Letter 1]

25 Nov 2019

Universal coverage but unmet need: national and regional estimates of attrition across the diabetes care continuum in Thailand

PONE-D-19-22038R1

Dear Dr. Stokes,

We are pleased to inform you that your manuscript has been judged scientifically suitable for publication and will be formally accepted for publication once it complies with all outstanding technical requirements.

With kind regards,

Nayu Ikeda, Ph.D.

Academic Editor

PLOS ONE

Additional Editor Comments (optional):

Reviewers' comments:

Reviewer's Responses to Questions

**Comments to the Author**

1. If the authors have adequately addressed your comments raised in a previous round of review and you feel that this manuscript is now acceptable for publication, you may indicate that here to bypass the “Comments to the Author” section, enter your conflict of interest statement in the “Confidential to Editor” section, and submit your "Accept" recommendation.

Reviewer #2: All comments have been addressed

2. Is the manuscript technically sound, and do the data support the conclusions?

Reviewer #2: Yes

3. Has the statistical analysis been performed appropriately and rigorously? 

Reviewer #2: (No Response)

4. Have the authors made all data underlying the findings in their manuscript fully available?

Reviewer #2: No

5. Is the manuscript presented in an intelligible fashion and written in standard English?

Reviewer #2: Yes

6. Review Comments to the Author

Reviewer #2: The authors have improved their manuscript. They have improved the tables presentation and the discussion. It could be published.

7. PLOS authors have the option to publish the peer review history of their article (what does this mean?). If published, this will include your full peer review and any attached files.

Reviewer #2: No

---

## [Editor Report · Acceptance letter]

2 Dec 2019

PONE-D-19-22038R1 

Universal coverage but unmet need: national and regional estimates of attrition across the diabetes care continuum in Thailand 

Dear Dr. Stokes:

I am pleased to inform you that your manuscript has been deemed suitable for publication in PLOS ONE. Congratulations! Your manuscript is now with our production department. 

With kind regards,

on behalf of

Dr. Nayu Ikeda 

Academic Editor

PLOS ONE